# Learning Wasserstein Embeddings

**Nicolas Courty**[*]
Université de Bretagne Sud
IRISA, UMR 6074, CNRS
`ncourty@irisa.fr`

**Rémi Flamary**
Université Côte d'Azur, OCA
Lagrange, UMR 7293, CNRS
`remi.flamary@unice.fr`

**Mélanie Ducoffe**
Université Côte d'Azur
I3S, UMR 7271, CNRS
`melanie.ducoffe@unice.fr`

## Abstract

The Wasserstein distance received a lot of attention recently in the community of machine learning, especially for its principled way of comparing distributions. It has found numerous applications in several hard problems, such as domain adaptation, dimensionality reduction or generative models. However, its use is still limited by a heavy computational cost. Our goal is to alleviate this problem by providing an approximation mechanism that allows to break its inherent complexity. It relies on the search of an embedding where the Euclidean distance mimics the Wasserstein distance. We show that such an embedding can be found with a siamese architecture associated with a decoder network that allows to move from the embedding space back to the original input space. Once this embedding has been found, computing optimization problems in the Wasserstein space (*e.g.* barycenters, principal directions or even archetypes) can be conducted extremely fast. Numerical experiments supporting this idea are conducted on image datasets, and show the wide potential benefits of our method.

## 1 Introduction

The Wasserstein distance is a powerful tool based on the theory of optimal transport to compare data distributions with wide applications in image processing, computer vision and machine learning (Kolouri et al., 2017). In a context of machine learning, it has recently found numerous applications, *e.g.* domain adaptation (Courty et al., 2017), or word embedding (Huang et al., 2016). In the context of deep learning, the Wasserstein appeared recently to be a powerful loss in generative models (Arjovsky et al., 2017) and in multi-label classification (Frogner et al., 2015). Its power comes from two major reasons: *i)* it allows to operate on empirical data distributions in a non-parametric way *ii)* the geometry of the underlying space can be leveraged to compare the distributions in a geometrically sound way. The space of probability measures equipped with the Wasserstein distance can be used to construct objects of interest such as barycenters (Agueh & Carlier, 2011) or geodesics (Seguy & Cuturi, 2015) that can be used in data analysis and mining tasks.

More formally, let $X$ be a metric space endowed with a metric $d_X$. Let $p \in (0, \infty)$ and $\mathcal{P}_p(X)$ the space of all Borel probability measures $\mu$ on $X$ with finite moments of order p, *i.e.* $\int_X d_X(x, x_0)^p d\mu(x) < \infty$ for all $x_0$ in $X$. The p-Wasserstein distance between $\mu$ and $\nu$ is defined as:

$$W_p(\mu, \nu) = \left( \inf_{\pi \in \Pi(\mu, \nu)} \iint_{X \times X} d(x, y)^p d\pi(x, y) \right)^{\frac{1}{p}}. \tag{1}$$

Here, $\Pi(\mu, \nu)$ is the set of probabilistic couplings $\pi$ on $(\mu, \nu)$. As such, for every Borel subsets $A \subseteq X$, we have that $\mu(A) = \pi(X \times A)$ and $\nu(A) = \pi(A \times X)$. It is well known that $W_p$ defines a metric over $\mathcal{P}_p(X)$ as long as $p \geq 1$ (e.g. (Villani, 2009), Definition 6.2).

---

[*]All three authors contributed equally

When $p = 1$, $W_1$ is also known as Earth Mover's distance (EMD) or Monge-Kantorovich distance. The geometry of $(\mathcal{P}_p(X), W_1(X))$ has been thoroughly studied, and there exists several works on computing EMD for point sets in $\mathbb{R}^k$ (e.g. Shirdhonkar & Jacobs (2008)). However, in a number of applications the use of $W_2$ (a.k.a *root mean square bipartite matching distance*) is a more natural distance arising in computer vision (Bonneel et al., 2015), computer graphics (Bonneel et al., 2011; de Goes et al., 2012; Solomon et al., 2015a; Bonneel et al., 2016) or machine learning (Cuturi & Doucet, 2014; Courty et al., 2017). See (de Goes et al., 2012) for a discussion on the quality comparison between $W_1$ and $W_2$.

Yet, the deployment of Wasserstein distances in a wide class of applications is somehow limited, especially because of an heavy computational burden. In the discrete version of the above optimisation problem, the number of variables scale quadratically with the number of samples in the distributions, and solving the associated linear program with network flow algorithms is known to have a cubical complexity. While recent strategies implying slicing technique (Bonneel et al., 2015; Kolouri et al., 2016a), entropic regularization (Cuturi, 2013; Benamou et al., 2015; Solomon et al., 2015b) or involving stochastic optimization (Genevay et al., 2016), have emerged, the cost of computing pairwise Wasserstein distances between a large number of distributions (like an image collection) is prohibitive. This is all the more true if one considers the problem of computing barycenters (Cuturi & Doucet, 2014; Benamou et al., 2015) or population means. A recent attempt by Staib and colleagues (Staib et al., 2017) use distributed computing for solving this problem in a scalable way.

We propose in this work to learn an Euclidean embedding of distributions where the Euclidean norm approximates the Wasserstein distances. Finding such an embedding enables the use of standard Euclidean methods in the embedded space and significant speedup in pairwise Wasserstein distance computation, or construction of objects of interests such as barycenters. The embedding is expressed as a deep neural network, and is learnt with a strategy similar to those of Siamese networks (Chopra et al., 2005). We also show that simultaneously learning the inverse of the embedding function is possible and allows for a reconstruction of a probability distribution from the embedding. We first start by describing existing works on Wasserstein space embedding. We then proceed by presenting our learning framework and give proof of concepts and empirical results on existing datasets.

## 2 RELATED WORK

**Metric embedding**    The question of metric embedding usually arises in the context of approximation algorithms. Generally speaking, one seeks a new representation (embedding) of data at hand in a new space where the distances from the original space are preserved. This new representation should, as a positive side effect, offers computational ease for time-consuming task (e.g. searching for a nearest neighbor), or interpretation facilities (e.g. visualization of high-dimensional datasets). More formally, given two metrics spaces $(X, d_X)$ and $(Y, d_y)$ and $D \in [1, \infty)$, a mapping $\phi : X \rightarrow Y$ is an embedding with distortion at most $D$ if there exists a coefficient $\alpha \in (0, \infty)$ such that $\alpha d_X(x, y) \leq d_Y(\phi(x), \phi(y)) \leq D\alpha d_X(x, y)$. Here, the $\alpha$ parameter is to be understood as a global scaling coefficient. The **distortion** of the mapping is the infimum over all possible $D$ such that the previous relation holds. Obviously, the lower the $D$, the better the quality of the embedding is. It should be noted that the existence of exact (**isometric**) embedding ($D = 1$) is not always guaranteed but sometimes possible. Finally, the embeddability of a metric space into another is possible if there exists a mapping with constant distortion. A good introduction on metric embedding can be found in (Matoušek, 2013).

**Theoretical results on Wasserstein space embedding**    Embedding Wasserstein space in normed metric space is still a theoretical and open questions (Matoušek & Naor, 2011). Most of the theoretical guarantees were obtained with $W_1$. In the simple case where $X = \mathbb{R}$, there exists an isometric embedding with $L_1$ between two absolutely continuous (*wrt.* the Lebesgue measure) probability measures $\mu$ and $\nu$ given by their by their cumulative distribution functions $F_\mu$ and $F_\nu$, *i.e.* $W_1(\mu, \nu) = \int_{\mathbb{R}} |F_\mu(x) - F_\nu(x)| dx$. This fact has been exploited in the computation of sliced Wasserstein distance (Bonneel et al., 2015; Kolouri et al., 2016c). Conversely, there is no known isometric embedding for pointsets in $[n]^k = \{1, 2, \ldots, n\}^k$, *i.e.* regularly sampled grids in $\mathbb{R}^k$, but best known distortions are between $O(k \log n)$ and $\Omega(k + \sqrt{\log n})$ (Charikar, 2002; Indyk & Thaper, 2003; Khot & Naor, 2006). Regarding $W_2$, recent results (Andoni et al., 2016) have shown there does not exist meaningful embedding over $\mathbb{R}^3$ with constant approximation. Their results show notably

that an embedding of pointsets of size $n$ into $L_1$ must incur a distortion of $O(\sqrt{\log n})$. Regarding our choice of $W_2^2$, there does not exist embeddability results up to our knowledge, but we show that, for a population of locally concentrated measures, a good approximation can be obtained with our technique. We now turn to existing methods that consider local linear approximations of the transport problem.

**Linearization of Wasserstein space**    Another line of work (Wang et al., 2013; Kolouri et al., 2016b) also considers the Riemannian structure of the Wasserstein space to provide meaningful linearization by projecting onto the tangent space. By doing so, they notably allows for faster computation of pairwise Wasserstein distances (only $N$ transport computations instead of $N(N-1)/2$ with $N$ the number of samples in the dataset) and allow for statistical analysis of the embedded data. They proceed by specifying a template element and compute, from particle approximations of the data, linear transport plans with this template element, that allow to derive an embedding used for analysis. Seguy and Cuturi (Seguy & Cuturi, 2015) also proposed a similar pipeline, based on velocity field, but without relying on an implicit embedding. It is to be noted that for data in 2D, such as images, the use of cumulative Radon transform also allows for an embedding which can be used for interpolation or analysis (Bonneel et al., 2015; Kolouri et al., 2016a), by exploiting the exact solution of the optimal transport in 1D through cumulative distribution functions.

Our work is the first to propose to learn a generic embedding rather than constructing it from explicit approximations/transformations of the data and analytical operators such as Riemannian Logarithm maps. As such, our formulation is generic and adapts to any type of data. Finally, since the mapping to the embedded space is constructed explicitly, handling unseen data does not require to compute new optimal transport plans or optimization, yielding extremely fast computation performances, with similar approximation performances.

## 3   DEEP WASSERSTEIN EMBEDDING (DWE)

### 3.1   WASSERSTEIN LEARNING AND RECONSTRUCTION WITH SIAMESE NETWORKS

We discuss here how our method, coined DWE for Deep Wasserstein Embedding, learns in a supervised way a new representation of the data. To this end we need a pre-computed dataset that consists of pairs of histograms $\{x_i^1, x_i^2\}_{i \in 1, \ldots, n}$ of dimensionality $d$ and their corresponding $W_2^2$ Wasserstein distance $\{y_i = W_2^2(x_i^1, x_i^2)\}_{i \in 1, \ldots, n}$. One immediate way to solve the problem would be to concatenate the samples $x^1$ and $x^2$ and learn a deep network that predicts $y$. This would work in theory but it would prevent us from interpreting the Wasserstein space and it is not by default symmetric which is a key property of the Wasserstein distance.

Another way to encode this symmetry and to have a meaningful embedding that can be used more broadly is to use a Siamese neural network (Bromley et al., 1994). Originally designed for metric learning purpose and similarity learning (based on labels), this type of architecture is usually defined by replicating a network which takes as input two samples from the same learning set, and learns a mapping to new space with a contrastive loss. It has mainly been used in computer vision, with successful applications to face recognition (Chopra et al., 2005) or one-shot learning for example (Koch et al., 2015). Though its capacity to learn meaningful embeddings has been highlighted in (Weston et al., 2012), it has never been used, to the best of our knowledge, for mimicking a specific distance that exhibits computation challenges. This is precisely our objective here.

We propose to learn and embedding network $\phi$ that takes as input a histogram and project it in a given Euclidean space of $\mathbb{R}^p$. In practice, this embedding should mirror the geometrical property of the Wasserstein space. We also propose to regularize the computation of this embedding by adding a reconstruction loss based on a decoding network $\psi$. This has two important impacts: First we observed empirically that it eases the learning of the embedding and improves the generalization performance of the network (see experimental results in appendix) by forcing the embedded representation to catch sufficient information of the input data to allow a good reconstruction. This type of autoencoder regularization loss has been discussed in (Yu et al., 2013) in the different context of embedding learning. Second, using a decoder network allows the interpretation of the results, which is of prime importance in several data-mining tasks (discussed in the next subsection).

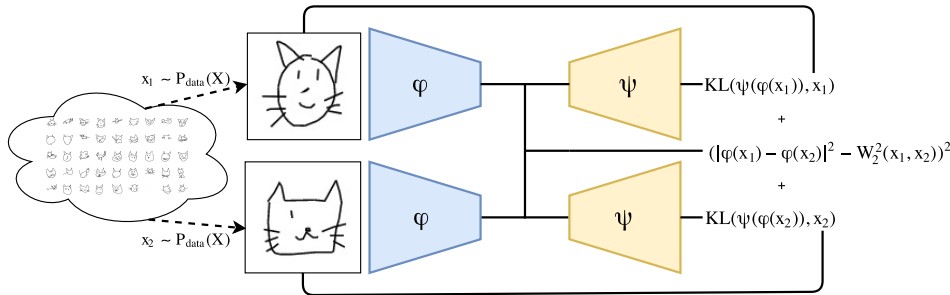

Figure 1: Architecture of the Deep Wasserstein Embedding: two samples are drawn from the data distribution and set as input of the same network ($\phi$) that computes the embedding. The embedding is learnt such that the squared Euclidean distance in the embedding mimics the Wasserstein distance. The embedded representation of the data is then decoded with a different network ($\psi$), trained with a Kullback-Leibler divergence loss.

An overall picture depicting the whole process is given in Figure 1. The global objective function reads

$$\min_{\phi,\psi} \quad \sum_i \left\| \|\phi(x_i^1) - \phi(x_i^2)\|^2 - y_i \right\|^2 + \lambda \sum_i \text{KL}(\psi(\phi(x_i^1)), x_i^1) + \text{KL}(\psi(\phi(x_i^2)), x_i^2) \quad (2)$$

where $\lambda > 0$ weights the two data fitting terms and $\text{KL}(,)$ is the Kullbach-Leibler divergence. This choice is motivated by the fact that the Wasserstein metric operates on probability distributions.

## 3.2 Wasserstein data mining in the embedded space

Once the functions $\phi$ and $\psi$ have been learned, several data mining tasks can be operated in the Wasserstein space. We discuss here the potential applications of our computational scheme and its wide range of applications on problems where the Wasserstein distance plays an important role. Though our method is not an exact Wasserstein estimator, we empirically show in the numerical experiments that it performs very well and competes favorably with other classical computation strategies.

**Wasserstein barycenters (Agueh & Carlier, 2011; Cuturi & Doucet, 2014; Bonneel et al., 2016).** Barycenters in Wasserstein space were first discussed by Agueh and Carlier (Agueh & Carlier, 2011). Designed through an analogy with barycenters in a Euclidean space, the Wasserstein barycenters of a family of measures are defined as minimizers of a weighted sum of squared Wasserstein distances. In our framework, barycenters can be obtained as

$$\bar{x} = \arg\min_x \sum_i \alpha_i W(x, x_i) \approx \psi(\sum_i \alpha_i \phi(x_i)), \quad (3)$$

where $x_i$ are the data samples and the weights $\alpha_i$ obeys the following constraints: $\sum_i \alpha_i = 1$ and $\alpha_i > 0$. Note that when we have only two samples, the barycenter corresponds to a Wasserstein interpolation between the two distributions with $\boldsymbol{\alpha} = [1 - t, t]$ and $0 \leq t \leq 1$ (Santambrogio, 2014). When the weights are uniform and the whole data collection is considered, the barycenter is the Wasserstein population mean, also known as Fréchet mean (Bigot et al., 2017).

**Principal Geodesic Analysis in Wasserstein space (Seguy & Cuturi, 2015; Bigot et al., 2017).** PGA, or Principal Geodesic Analysis, has first been introduced by Fletcher et al. (Fletcher et al., 2004). It can be seen as a generalization of PCA on general Riemannian manifolds. Its goal is to find a set of directions, called geodesic directions or principal geodesics, that best encode the statistical variability of the data. It is possible to define PGA by making an analogy with PCA. Let $x_i \in \mathbb{R}^n$ be a set of elements, the classical PCA amounts to *i)* find $\bar{x}$ the mean of the data and subtract it to all the samples *ii)* build recursively a subspace $V_k = \text{span}(v_1, \cdots, v_k)$ by solving the following maximization problem:

$$v_1 = \text{argmax}_{|v|=1} \sum_{i=1}^n (v.x_i)^2, \quad v_k = \text{argmax}_{|v|=1} \sum_{i=1}^n \left( (v.x_i)^2 + \sum_{j=1}^{k-1} (v_j.x_i)^2 \right). \quad (4)$$

Fletcher gives a generalization of this problem for complete geodesic spaces by extending three important concepts: **variance** as the expected value of the squared Riemannian distance from mean, **Geodesic subspaces** as a portion of the manifold generated by principal directions, and a **projection** operator onto that geodesic submanifold. The space of probability distribution equipped with the Wasserstein metric $(\mathcal{P}_p(X), W_2^2(X))$ defines a geodesic space with a Riemannian structure (Santambrogio, 2014), and an application of PGA is then an appealing tool for analyzing distributional data. However, as noted in (Seguy & Cuturi, 2015; Bigot et al., 2017), a direct application of Fletcher's original algorithm is intractable because $\mathcal{P}_p(X)$ is infinite dimensional and there is no analytical expression for the exponential or logarithmic maps allowing to travel to and from the corresponding Wasserstein tangent space. We propose a novel PGA approximation as the following procedure: *i)* find $\overline{x}$ the approximate Fréchet mean of the data as $\overline{x} = \frac{1}{N}\sum_i^N \phi(x_i)$ and subtract it to all the samples *ii)* build recursively a subspace $V_k = \mathrm{span}(v_1, \cdots, v_k)$ in the embedding space ($v_i$ being of the dimension of the embedded space) by solving the following maximization problem:

$$v_1 = \mathrm{argmax}_{|v|=1} \sum_{i=1}^n (v.\phi(x_i))^2, \quad v_k = \mathrm{argmax}_{|v|=1} \sum_{i=1}^n \left( (v.\phi(x_i))^2 + \sum_{j=1}^{k-1}(v_j.\phi(x_i))^2 \right). \quad (5)$$

which is strictly equivalent to perform PCA in the embedded space. Any reconstruction from the corresponding subspace to the original space is conducted through $\psi$. We postpone a detailed analytical study of this approximation to subsequent works, as it is beyond the goals of this paper.

**Other possible methods.** As a matter of facts, several other methods that operate on distributions can benefit from our approximation scheme. Most of those methods are the transposition of their Euclidian counterparts in the embedding space. Among them, clustering methods, such as Wasserstein k-means (Cuturi & Doucet, 2014), are readily adaptable to our framework. Recent works have also highlighted the success of using Wasserstein distance in dictionary learning (Rolet et al., 2016) or archetypal Analysis (Wu & Tabak, 2017).

## 4    NUMERICAL EXPERIMENTS

In this section we evaluate the performances of our method on grayscale images normalized as histograms. Images are offering a nice testbed because of their dimensionality and because large datasets are frequently available in computer vision.

### 4.1    ARCHITECTURE FOR DWE BETWEEN GRAYSCALE IMAGES

The framework of our approach as shown in Fig 1 consists of an encoder $\phi$ and a decoder $\psi$ composed as a cascade. The encoder produces the representation of input images $h = \phi(x)$. The architecture used for the embedding $\phi$ consists in 2 convolutional layers with ReLU activations: first a convolutional layer of 20 filters with a kernel of size 3 by 3, then a convolutional layer of 5 filters of size 5 by 5. The convolutional layers are followed by two linear dense layers respectively of size 100 and the final layer of size $p = 50$. The architecture for the reconstruction $\psi$ consists in a dense layer of output 100 with ReLU activation, followed by a dense layer of output 5*784. We reshape the layer to map the input of a convolutional layer: the output vector is (5,28,28) 3D-tensor. Eventually, we invert the convolutional layers of $\phi$ with two convolutional layers: first a convolutional layer of 20 filters with ReLU activation and a kernel of size 5 by 5, followed by a second layer with 1 filter, with a kernel of size 3 by 3. Eventually the decoder outputs a reconstruction image of shape 28 by 28. In this work, we only consider grayscale images, that are normalized to represent probability distributions. Hence each image is depicted as an histogram. In order to normalize the decoder reconstruction we use a softmax activation for the last layer.

All the dataset considered are handwritten data and hence holds an inherent sparsity. In our case, we cannot promote the output sparsity through a convex L1 regularization because the softmax outputs positive values only and forces the sum of the output to be 1. Instead, we apply a $\ell_p^p$ pseudo -norm regularization with $p = 1/2$ on the reconstructed image, which promotes sparse output and allows for a sharper reconstruction of the images (Gasso et al., 2009).

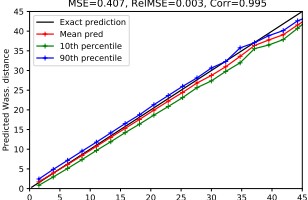

| Method | $W_2^2$/sec |
|---|---|
| LP network flow (1 CPU) | 192 |
| DWE Indep. (1 CPU) | 3 633 |
| DWE Pairwise (1 CPU) | 213 384 |
| DWE Indep. (GPU) | 233 981 |
| DWE Pairwise (GPU) | 10 477 901 |

Figure 2: Prediction performance on the MNIST dataset. (Figure) The test performance are as follows: MSE=0.41, Relative MSE=0.003 and Correlation=0.995. (Table) Computational performance of $W_2^2$ and DWE given as average number of $W_2^2$ computation per seconds for different configurations.

## 4.2 MNIST DIGIT DATASET

**Dataset and training.** Our first numerical experiment is performed on the well known MNIST digits dataset. This dataset contains $28 \times 28$ images from 10 digit classes In order to create the training dataset we draw randomly one million pairs of indexes from the 60 000 training samples and compute the exact Wasserstein distance with a squared Euclidean ground metric using the POT toolbox (Flamary & Courty, 2017). All those pairwise distances can be computed in an embarrassingly parallel scheme (1h30 on 1 CPU). Among this million, 700 000 are used for learning the neural network, 200 000 are used for validation and 100 000 pairs are used for testing purposes. The DWE model is learnt on a GTX TitanX Maxwell 980 GPU node and takes around 1h20 with a stopping criterion computed from on a validation set.

**Numerical precision and computational performance** The true and predicted values for the Wasserstein distances are given in Fig. 2. We can see that we reach a good precision with a test MSE of 0.4 and a relative MSE of 2e-3. The correlation is of 0.996 and the quantiles show that we have a very small uncertainty with only a slight bias for large values where only a small number of samples is available. This results show that a good approximation of the $W_2^2$ can be performed by our approach ($\approx$1e-3 relative error).

Now we investigate the ability of our approach to compute $W_2^2$ efficiently. To this end we compute the average speed of Wasserstein distance computation on test dataset to estimate the number of $W_2^2$ computations per second in the Table of Fig. 2. Note that there are 2 ways to compute the $W_2^2$ with our approach denoted as Indep and Pairwise. This comes from the fact that our $W_2^2$ computation is basically a squared Euclidean norm in the embedding space. The first computation measures the time to compute the $W_2^2$ between independent samples by projecting both in the embedding and computing their distance. The second computation aims at computing all the pairwise $W_2^2$ between two sets of samples and this time one only needs to project the samples once and compute all the pairwise distances, making it more efficient. Note that the second approach would be the one used in a retrieval problem where one would just embed the query and then compute the distance to all or a selection of the dataset to find a Wasserstein nearest neighbor for instance. The speedup achieved by our method is very impressive even on CPU with speedup of x18 and x1000 respectively for Indep and Pairwise. But the GPU allows an even larger speedup of respectively x1000 and x500 000 with respect to a state-of-the-art C compiled Network Flow LP solver of the POT Toolbox (Flamary & Courty, 2017; Bonneel et al., 2011). Of course this speed-up comes at the price of a time-consuming learning phase, which makes our method better suited for mining large scale datasets and online applications.

**Wasserstein Barycenters** Next we evaluate our embedding on the task of computing Wasserstein Barycenters for each class of the MNIST dataset. We take 1000 samples per class from the test dataset and compute their uniform weight Wasserstein Barycenter using Eq. 3. The resulting barycenters and their Euclidean means are reported in Fig. 3. Note that not only those barycenters are sensible but also conserve most of their sharpness which is a problem that occurs for regularized barycenters (Solomon et al., 2015b; Benamou et al., 2015). The computation of those barycenters is also very efficient since it requires only 20ms per barycenter (for 1000 samples) and its complexity scales linearly with the number of samples.

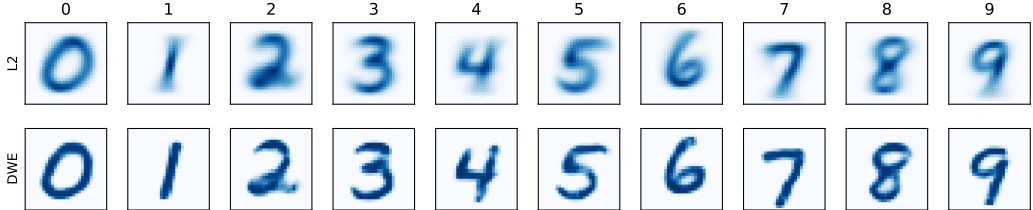

Figure 3: Barycenter estimation on each class of the MNIST dataset for squared Euclidean distance (L2) and Deep Wasserstein Embedding (DWE).

| Class 0 | | | | | | Class 1 | | | | | | Class 4 | | | | | |
| L2 | | | DWE | | | L2 | | | DWE | | | L2 | | | DWE | | |
| 1 | 2 | 3 | 1 | 2 | 3 | 1 | 2 | 3 | 1 | 2 | 3 | 1 | 2 | 3 | 1 | 2 | 3 |

Figure 4: Principal Geodesic Analysis for classes 0,1 and 4 from the MNIST dataset for squared Euclidean distance (L2) and Deep Wasserstein Embedding (DWE). For each class and method we show the variation from the barycenter along one of the first 3 principal modes of variation.

**Principal Geodesic Analysis**    We report in Figure 4 the Principal Component Analysis (L2) and Principal Geodesic Analysis (DWE) for 3 classes of the MNIST dataset. We can see that using Wasserstein to encode the displacement of mass leads to more semantic and nonlinear subspaces such as rotation/width of the stroke and global sizes of the digits. This is well known and has been illustrated in (Seguy & Cuturi, 2015). Nevertheless our method allows for estimating the principal component even in large scale datasets and our reconstruction seems to be more detailed compared to (Seguy & Cuturi, 2015) maybe because our approach can use a very large number of samples for subspace estimation.

### 4.3    GOOGLE DOODLE DATASET

**Datasets**    The Google Doodle dataset is a crowd sourced dataset that is freely available from the web[1] and contains 50 million drawings. The data has been collected by asking users to hand draw with a mouse a given object or animal in less than 20 seconds. This lead to a large number of examples for each class but also a lot of noise in the sens that people often get stopped before the end of their drawing .We used the numpy bitmaps format proposed on the quick draw github account. Those are made of the simplified drawings rendered into 28x28 grayscale images. These images are aligned to the center of the drawing's bounding box. In this paper we downloaded the classes Cat, Crab and Faces and tried to learn a Wasserstein embedding for each of these classes with the same architecture as used for MNIST. In order to create the training dataset we draw randomly 1 million pairs of indexes from the training samples of each categories and compute the exact Wasserstein distance with squared Euclidean ground metric using the POT toolbox (Flamary & Courty, 2017). Same as for MNIST, 700 000 are used for learning the neural network, 200 000 are used for validation

---

[1]https://quickdraw.withgoogle.com/data

| Learn \ Test | CAT | CRAB | FACE | MNIST | | Learn \ Test | CAT | CRAB | FACE | MNIST |
|---|---|---|---|---|---|---|---|---|---|---|
| CAT | **1.491** | *1.818* | 1.927 | 12.525 | | CAT | **0.004** | 0.007 | 0.011 | 0.082 |
| CRAB | *2.679* | **0.918** | 3.510 | 11.750 | | CRAB | 0.009 | **0.004** | 0.018 | 0.075 |
| FACE | *4.884* | 4.843 | **1.313** | 52.994 | | FACE | 0.018 | 0.024 | **0.008** | 0.329 |
| MNIST | 9.776 | 6.689 | *4.387* | **0.407** | | MNIST | 0.028 | 0.030 | 0.026 | **0.003** |

| (a) MSE | (b) Relative MSE |
|---|---|

Table 1: Cross performance between the DWE embedding learned on each datasets. On each row, we observe the MSE (table a) and relative MSE (table b) on the test set of each dataset given a DWL (Cat, Crab, Faces and MNIST).

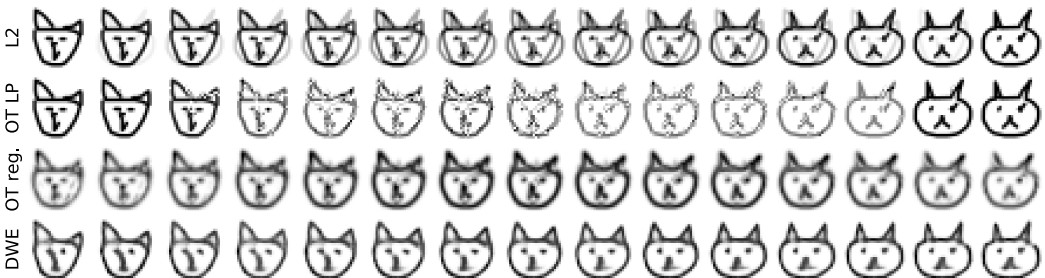

Figure 5: Comparison of the interpolation with L2 Euclidean distance (top), LP Wasserstein interpolation (top middle) regularized Wasserstein Barycenter (down middle) and DWE (down).

and 100 000 pairs are used for testing purposes. Each of the three categories (Cat, Crab and Faces) holds respectively 123 202, 126 930 and 161 666 training samples.

**Numerical precision and cross dataset comparison** The numerical performances of the learned models on each of the doodle dataset is reported in the diagonal of Table 1. Those datasets are much more difficult than MNIST because they have not been curated and contain a very large variance due to numerous unfinished doodles. An interesting comparison is the cross comparison between datasets where we use the embedding learned on one dataset to compute the $W_2^2$ on another. The cross performances is given in Table 1 and shows that while there is definitively a loss in accuracy of the prediction, this loss is limited between the doodle datasets that all have an important variety. Performance loss across doodle and MNIST dataset is larger because the latter is highly structured and one needs to have a representative dataset to generalize well which is not the case with MNIST. This also clearly highlights that our method finds a data-dependent embedding that is specific to the geometry of the learning set.

**Wasserstein interpolation** Next we qualitatively evaluate the subspace learned by DWE by comparing the Wasserstein interpolation of our approach with the true Wasserstein interpolation estimated by solving the OT linear program and by using regularized OT with Bregman projections (Benamou et al., 2015). The interpolation results for all those methods and the Euclidean interpolation are available in Fig. 5. The LP solver takes a long time (20 sec/interp) and leads to a "noisy" interpolation as already explained in (Cuturi & Peyré, 2016). The regularized Wasserstein barycenter is obtained more rapidly (4 sec/interp) but is also very smooth at the risk of loosing some details, despite choosing a small regularization that prevents numerical problems. Our reconstruction also looses some details due to the Auto-Encoder error but is very fast and can be done in real time (4 ms/interp).

## 5    CONCLUSION AND DISCUSSION

In this work we presented a computational approximation of the Wasserstein distance suitable for large scale data mining tasks. Our method finds an embedding of the samples in a space where the Euclidean distance emulates the behavior of the Wasserstein distance. Thanks to this embedding, numerous data analysis tasks can be conducted at a very cheap computational price. We forecast that this strategy can help in generalizing the use of Wasserstein distance in numerous applications.

However, while our method is very appealing in practice it still raises a few questions about the theoretical guarantees and approximation quality. First it is difficult to foresee from a given network architecture if it is sufficiently (or too much) complex for finding a successful embedding. It can be conjectured that it is dependent on the complexity of the data at hand and also the locality of the manifold where the data live in. Second, the theoretical existence results on such Wasserstein embedding with constant distortion are still lacking. Future works will consider these questions as well as applications of our approximation strategy on a wider range of ground loss and data mining tasks. Also, we will study the transferability of one database to another (i.e. leveraging on previously computed embedding) to diminish the computational burden of computing Wasserstein distances on numerous pairs for the learning process, by considering for instance domain adaptation strategies between embeddings.

## ACKNOWLEDGEMENTS

This work benefited from the support of the project OATMIL ANR-17-CE23-0012 of the French National Research Agency (ANR), and from using Inria Sophia Antipolis - Mediterranée computation cluster Nef. The authors wish to also thank Romain Tavenard for discussions on the subject.

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

## A    EFFECT ON USING AN AUTOENCODER LOSS IN THE LEARNING PROCESS

We discuss here the role of the decoder, not only as a matter of interpreting the results, but rather as a regularizer. We train our DWE on MNIST with and without the decoder and compares the learning curves of the MSE on the validation set. In Figure 6, DWE achieves a lower MSE with the decoder, which enforces the use of a decoder into our framework.

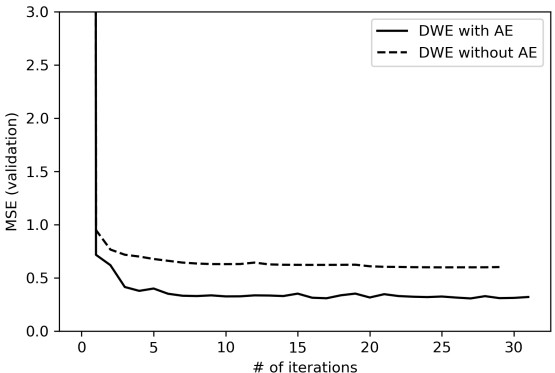

Figure 6: $W_2^2$ validation MSE along the number of epochs for the MNIST dataset (DWE).

## B    COMPLEMENTARY RESULTS ON GOOGLE DOODLE DATASET

We illustrate here the plurality of examples found in this dataset by drawing random excerpts in Fig. 7. There exist also a lot of outlier images (scribblings, texts, etc.). As discussed in the main text several drawings are unfinished and/or do not represent correctly the required class.

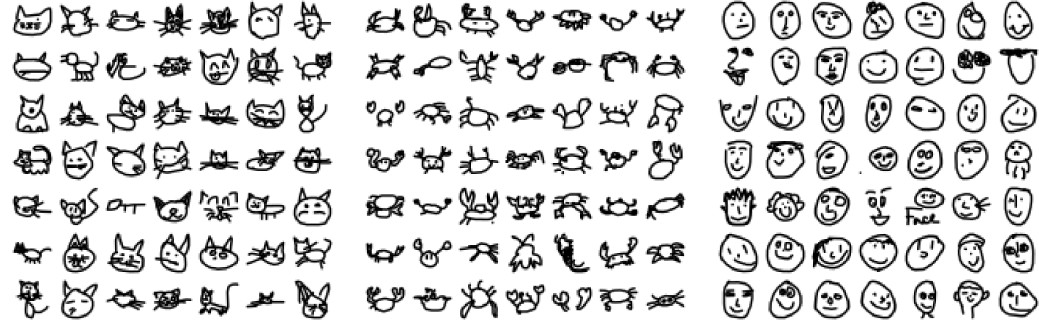

Figure 7: Examples drawn from the 3 Google Doodle datasets (left) cat dataset, (center) Crab dataset (right) Face dataset.

We then compute the Wasserstein interpolation between four samples of each datasets in Fig. 8. Note that these interpolation might not be optimal w.r.t. the objects but we clearly see a continuous displacement of mass that is characteristic of optimal transport. This leads to surprising artefacts for example when the eye of a face fuse with the border while the nose turns into an eye. Also note that there is no reason for a Wasserstein barycenter to be a realistic sample.

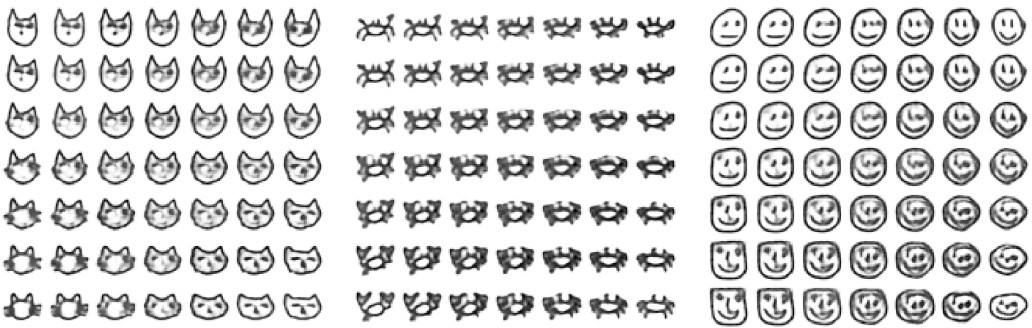

Figure 8: Interpolation between four samples of each datasets using DWE. (left) cat dataset, (center) Crab dataset (right) Face dataset.

In Fig. 9 we show the quantitative evaluation for DWE on the three datasets, that correspond to Table 1 in the paper. The reported MSE performances correspond to the ones in the diagonal of Table 1. We can see that the deviation is larger for large values of $W_2^2$ mainly because of the small number of training samples for those values.

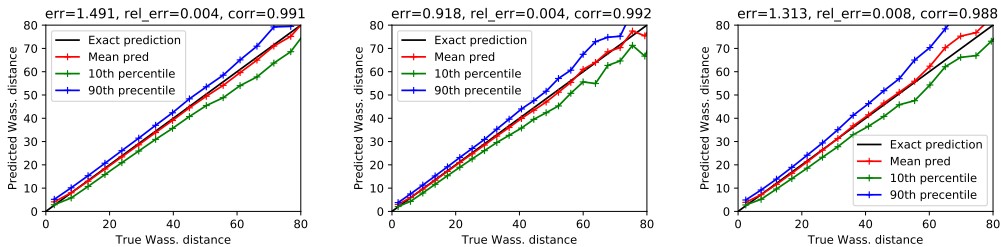

Figure 9: Performance of DWE on the Cat (left) Crab (center) and Face (right) doodle datsets.

We report in Fig. 10 a nearest neighbor walk (sequential jumps to the nearest, in the sense of the considered metric, image that has not already been seen) on a subset of 10000 test samples starting with the same image but using either the L2 distance in the input or DWE embedded space. Note that the L2 in input space is here very sensible to outliers (black squares) that are rare in the dataset but

have a L2 distance rather small to all other examples (most sequences converge to those samples). Conversely the DWE neighbors follow a smooth trajectory along the examples. This illustrates the advantage of $W_2^2$ for image retrieval, which is made computationally possible with DWE.

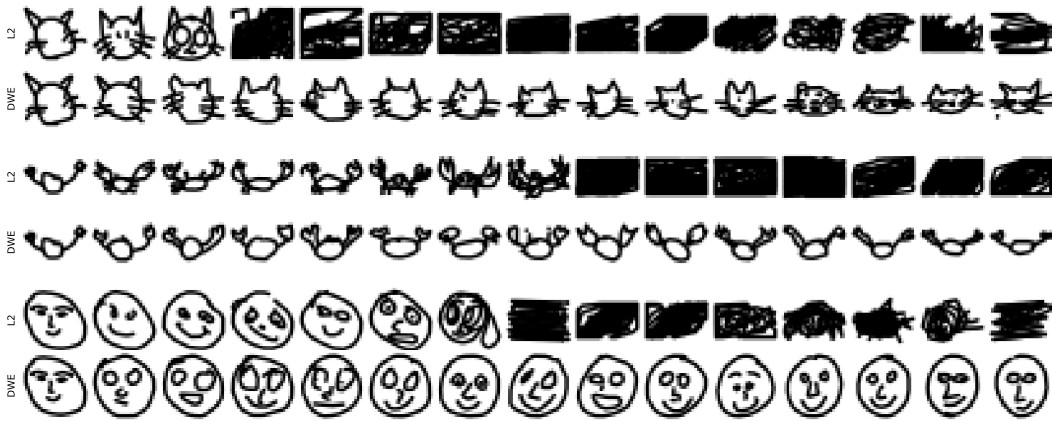

Figure 10: Nearest neighbor walk along the 3 datasets when using L2 or DWE for specifying the neighborhood. (up) Cat dataset, (middle) Crab dataset (down) Face dataset.

