# OpenReview forum: "Learning Wasserstein Embeddings"
_ICLR.cc/2018/Conference — Accept (Poster)_

### Official Review · AnonReviewer1 · 2017-11-29
**An efficient approach to compute Wasserstein distances and to perform various related analyses.**

**Rating:** 7
**Confidence:** 3

**Review:**

The paper proposes to use a deep neural network to embed probability distributions in a vector space, where the Euclidean distance in that space matches the Wasserstein distance in the original space of probability distributions. A dataset of pairs of probability distributions and their Wasserstein distance is collected, and serves as a target to be predicted by the deep network.

The method is straightforward, and clearly explained. Two analyses based on Wasserstein distances (computing barycenters, and performing geodesic analysis) are then performed directly in the embedded space.

The authors claim that the proposed method produces sharper barycenters than those learned using the standard (smooth) Wasserstein distance. It is unclear from the paper whether the advantage comes from the ability of the method to scale better and use more examples, or to be able to use the non-smooth Wasserstein distance, or finally, whether the learning of a deep embedding yields improved extrapolation properties. A short discussion could be added. It would also be interesting to provide some guidance on what is a good structure for the encoder (e.g. should it include spatial pooling layers?)

The term “Wasserstein deep learning” is probably too broad, “deep Wasserstein embedding” could be more appropriate.

The last line of future work in the conclusion seems to describe the experiment of Table 1.

---

> ### Author Response · Authors · 2017-12-21
> **Regarding the last line of conclusion**
>
> Indeed the first of line of future work is concerned with transferability issue of a learned mapping toward a new
> dataset. In the paper we have examined if the mapping was transferable and we observed that it is mostly data dependent. In a future line of work, we would like to see if we can ‘transfer’ an already learnt embedding to work on a different dataset (as would work a domain adaptation technique). We have rephrased the text to state this idea more clearly.

---

> ### Author Response · Authors · 2017-12-21
> **Regarding guidance for a good structure of the encoder**
>
> This is a difficult question. The Wasserstein distance cares about spatial location, hence adding spatial pooling in our network may coarser the embedding.  For bigger images, we may consider strided convolutions instead of max-pooling. This is currently under examination as we are working with larger images, but with no definitive answer for the moment.

---

> ### Author Response · Authors · 2017-12-21
> **Regarding the quality of interpolation**
>
> The main interest of the method is to be able to compute a fast and accurate approximation of the true Wasserstein distance (and not the regularized one), but the embedding could also be learned to reflect a regularized version of W if needed by the application. The sharper quality of barycenters mostly comes with the fact that we are handling  true Wasserstein distances and not regularized ones

---

### Official Review · AnonReviewer2 · 2017-11-30
**Simple idea that is potentially useful in practice.**

**Rating:** 7
**Confidence:** 3

**Review:**

The paper presents a simple idea to reduce the computational cost of computing Wasserstein distance between a pair of histograms. Specifically, the paper proposes learning an embedding on the original histograms into a new space where Euclidean distance in the latter relates to the Wasserstein distance in the original space. Despite simplicity of the idea, I think it can potentially be useful practical tool, as it allows for very fast approximation of Wasserstein distance. The empirical results show that embeddings learned by the proposed model indeed provide a good approximation to the actual Wasserstein distances.

The paper is well-written and is easy to follow and understand. There are some grammar/spelling issues that can be fixed by a careful proofreading. Overall, I find the paper simple and interesting.

My biggest concern however is the applicability of this approach to high-dimensional data. The experiments in the paper are performed on 2D histograms (images). However, the number of cells in the histogram grows exponentially in dimension. This may turn this approach impractical even in a moderate-sized dimensionality, because the input to the learning scheme  requires explicit representation of the histogram, and the proposed method may quickly run into memory problems. In contrast, if one uses the non-learning based approach (standard LP formulation of Wasserstein distance), at least in case of W_1, one can avoid memory issues caused by the dimensionality by switching to the dual form of the LP. I believe that is an important property that has made computation of Wasserstein distance practical in high dimensional settings, but seems inapplicable to the learning scheme. If there is a workaround, please specify.

---

> ### Author Response · Authors · 2017-12-21
> **Regarding high dimensional settings**
>
> Indeed we agree with the reviewer that the input dimension of our embedding network scales linearly in terms of bins in the histograms. Note however that dual (or semi-dual) approaches require the computation of Kantorovich potentials that are scalar functions of the dimension of ambient (input) space, that turns to be of same size as the number of bins of the histogram. Hence both views require to process the data through networks that have the same input size and might suffer from the same problem of high dimensionality. If considering 2D, 3D or 4D tensors, note however that neural networks architecture are known to accommodate well to such dimensions (generally through convolution and pooling layers). We also note that in high dimensions, even computing a single Wasserstein distance is difficult, and a recent analysis [1] shows also the impact of dimensionality in estimating accurately the Wasserstein distance.
>
> [1] J. Weed, F. Bach. Sharp asymptotic and finite-sample rates of convergence of empirical measures in Wasserstein distance. Technical Report, Arxiv-1707.00087, 2017

---

### Official Review · AnonReviewer3 · 2017-11-30
**A clearly written, novel, straightforward and practical approach to Wasserstein distance--based image embeddings.**

**Rating:** 7
**Confidence:** 4

**Review:**

This paper proposes approximating the Wasserstein distance between normalized greyscale images based on a learnable approximately isometric embedding of images into Euclidean space. The paper is well written with clear and generally thorough prose. It presents a novel, straightforward and practical solution to efficiently computing Wasserstein distances and performing related image manipulations.

Major comments:

It sounds like the same image may be present in the training set and eval set. This is methodologically suspect, since the embedding may well work better for images seen during training. This affects all experimental results.

I was pleased to see a comparison between using exact and approximate Wasserstein distances for image manipulation in Figure 5, since that's a crucial aspect of whether the method is useful in practice. However the exact computation (OT LP) appears to be quite poor. Please explain why the approximation is better than the exact Wasserstein difference for interpolation. Relatedly, please summarize the argument in Cuturi and Peyre that is cited ("as already explained in").

Minor comments:

In section 3.1 and 4.1, "histogram" is used to mean normalized-to-sum-to-1 images, which is not the conventional meaning.

It would help to pick one of "Wasserstein Deep Learning" and "Deep Wasserstein Embedding" and use it and the acronym consistently throughout.

"Disposing of a decoder network" in section 3.1 should be "using a decoder network"?

In section 4.1, the architectural details could be clarified. What size are the input images? What type of padding for the convolutions? Was there any reason behind the chosen architecture? In particular the use of a dense layers followed by convolutional layers seems peculiar.

It would be helpful to say explicitly what "quadratic ground metric" means (i.e. W_2, I presume) in section 4.2 and elsewhere.

It would be helpful to give a sense of scale for the numbers in Table 1, e.g. give the 95th percentile Wasserstein distance. Perhaps use the L2 distance passed through a 1D-to-1D learned warping as a baseline.

Mention that OT stands for optimal transport in section 4.3.

Suggest mentioning "there is no reason for a Wasserstein barycenter to be a realistic sample" in the main text when first discussing barycenters.

---

> ### Author Response · Authors · 2017-12-20
> **Regarding presence of images of the learning sets in the testing set**
>
> With our settings, it may be possible to have some redundancy between the training and the test set. However, we ensure that statistically, it is highly unlikely to have redundant couples of images between the training and test set. Eventually, the higher the number of images to compute pairwise Wasserstein distance is, the lower is the probability of sharing images between the training and test set: especially when N > sqrt(100,000). We ensure this condition for every dataset ( N(mnist)=50000, N(face)=161666, N(crab)= 126930, N(cat)= 123202).
> Regarding our experiments on Principal Geodesic Analysis and Barycenter’s estimation, those have been done on test images independent from the training set.
>
> However, to clear any doubt regarding the efficiency of our method, we update Figure 2, Figure 9 and Table 1: we tested the pairwise Wasserstein distance with test images independent from the training set. Our results remain almost unchanged.

---

> ### Author Response · Authors · 2017-12-20
> **Regarding  OT LP results in Figure 5**
>
> We are referring to Figures 3.1 and 3.2 in the paper ‘A smoothed dual approach for variational Wasserstein problems’ from Cuturi and Peyré, that show how the exact solution of the linear program corresponding to an interpolation in the Wasserstein sense of two Gaussians  can lead to a staircase effect in the interpolated Gaussian, that is mainly due to discretization. We believe that the reconstructed images in our case suffer from the same discretization effect.

---

> ### Author Response · Authors · 2017-12-20
> **More information on the architecture used in the paper**
>
> We provide details about the architecture we used :
>
> Encoder :
> - input size (1, 28, 28)
> - a convolutional layer: 20 filters of kernel size 3 by 3, with zero padding and ReLu activation
> - a convolutional layer: 10 filters of kernel size 3 by 3, with zero padding and ReLu activation
> - a convolutional layer:5 filters of kernel size 5 by 5, with zero padding and ReLu activation
> - a fully connected layer with 100 output neurons and ReLu activation
> - a fully connected layer with 50 output neurons, Relu activation. The output is our embedding.
>
> Decoder :
> - input size (50,)
> - a fully connected layer with 100 output neurons and ReLu activation
> - a fully connected layer with 5*28*28 output neurons and ReLu activation
> - a reshape layer of target size (5, 28, 28)
> - a convolutional layer: 10 filters of kernel size 5 by 5, with zero padding and ReLu activation
> - a convolutional layer: 20 filters of kernel size 3 by 3, with zero padding and ReLu activation
> - a convolutional layer: 1 filter of kernel size 3 by 3, with zero padding and ReLu activation
> - a Softmax layer whose output is the image reconstruction.
>
> All weights are initialized with Glorot’s rule.
> In the encoder, there is no dense layer followed by a convolutional layer. However without max-pooling, we need dense layers at the end of the encoder to control the size of the embedding. Hence to mimic the inversion of each layer of the encoder, we indeed add dense layers followed by convolutional layers.
> We also plan to publish a version of our code on GitHub.

---

> ### Author Response · Authors · 2017-12-20
> **Some elements of answers wrt. minor comments**
>
> First of all, thanks for the reviewer for helping us to improve our manuscript.
>
> Regarding the Quadratic ground metric, it refers to squared Euclidean distance. As it was indeed not clear in the paper, we changed this notation in the revised version.
>
> Regarding the scale of table 1, the theoretical maximum distance is 1458 (all mass between pixels in opposite corners), in average the pairwise wasserstein distance if of the order 12 for MNIST and 15 for CAT, CRAB and FACES but with relative MSE of order 1e-3 (see Table 1 for the exact values) for which is quite large with respect to the quadratic mean error reported in the tables.

---

### Public Comment · ~Jianbo_Ye1 · 2017-12-06
**An interesting paper, and some questions**

I think it is an interesting paper for approximating OT on low-dimensional space.

Could the author comment on how accurate/applicable this approach will generalize to high dimensional distributions?


Also we know the closed formula for computing Wasserstein distance between multivariate Gaussians. Maybe a sanity check with Gaussians can strengthen the related claims numerically.

Thanks!

---

> ### Author Response · Authors · 2017-12-20
> **Some elements of response**
>
> Thanks for your comments. When referring to dimensionality of distributions, several dimensions can be taken into account: dimension of the ambient space, dimension of discretization (number of bins in the histograms) in a Eulerian setting or number of Diracs in a Lagrangian view of empirical distributions. Our method is for now adapted mostly to distributions with fixed discretization on a constant Eulerian grid, that corresponds to the input size of the embedding network. As such, it is difficult to consider empirical distributions that we would draw from multivariate Gaussians (hence with known and computable Wasserstein distance). Note that also in this case, a sampling error should be taken into account (and the exact W distance would be different from the theoretical one). We have started working on ways to embed empirical distributions in a similar framework as the one   developed in our paper but this is somehow out the scope of the proposed work. Regarding the generalization of our approach to larger number of bins in the histogram (Eulerian view), the problem of computing even a single Wasserstein distance may arise, especially because the size of the coupling scales quadratically in the number of bins. While for 2D and 3D histograms convolutional Wasserstein distances can be used to compute efficiently the Wasserstein distance, scaling to larger dimension of ambient space is still an open issue. Working with stochastic semi-dual or dual approaches such as in [Genevay et al. 2016] is a possible option, but it comes with higher computational costs, that prevents computing Wasserstein distances for a large number of pairs.

---

### Decision · Program_Chairs · 2018-01-29
**ICLR 2018 Conference Acceptance Decision**

**Decision:**

Accept (Poster)

**Comment:**

The paper presents a practical approach to compute Wasserstein distance based image embeddings. The Euclidean distance in the embedded space approximates the true Wasserstein distance, thus reducing the high computation cost associated with the latter.

Pros:
- Reviewers agree that the proposed solution is novel, straightforward and well described.
- Experiment demonstrate the usefulness of such embeddings for data mining tasks such as fast computation of barycenters & geodesic analysis.

Cons:
- Though the empirical analysis is convincing, the paper lacks theoretical analysis of the approximation quality.